# Comparative Gravimetric Studies on Carbon Steel Corrosion in Selected Fruit Juices and Acidic Chloride Media (HCl) at Different pH

**DOI:** 10.3390/ma14164755

**Published:** 2021-08-23

**Authors:** Stanley Udochukwu Ofoegbu

**Affiliations:** 1Centre for Mechanical Technology and Automation (TEMA), Department of Mechanical Engineering, Campus Universitário de Santiago, University of Aveiro, 3810-193 Aveiro, Portugal; ofoegbu.stanley@ua.pt; 2Department of Materials and Ceramic Engineering, CICECO-Aveiro Institute of Materials, Campus Universitário de Santiago, University of Aveiro, 3810-193 Aveiro, Portugal

**Keywords:** carbon steel, organic acids, antioxidants, corrosion products, iron oxides, microspheres contamination, food media, pH, solubility

## Abstract

Food contamination due to metal corrosion and the consequent leakage of metals into foods is a problem. Understanding the mechanism(s) of metal corrosion in food media is vital to evaluating, mitigating, and predicting contamination levels. Fruit juices have been employed as model corrosive media to study the corrosion behaviour of metallic material in food media. Carbon steel corrosion in fresh juices of tomato, orange, pineapple, and lemon, as well as dilute hydrochloric acid solutions at varied pH, was studied using scanning electron microscopy, gravimetric and spectrophotometric techniques, and comparisons made between the corrosivity of these juices and mineral acids of comparable pH. The corrosion of carbon steel in fruit juices and HCl solutions manifests as a combination of uniform and pitting corrosion. Gravimetric data acquired after one hour of immersion at ambient temperature (22 °C) indicated corrosion rates of 0.86 mm yr^−1^ in tomato juice (pH ≈ 4.24), 1.81 mm yr^−1^ in pineapple juice (pH ≈ 3.94), 1.52 mm yr^−1^ in orange juice (pH ≈ 3.58), and 2.89 mm yr^−1^ in lemon juice (pH ≈ 2.22), compared to 2.19 mm yr^−1^ in 10^−2^ M HCl (pH ≈ 2.04), 0.38 mm yr^−1^ in 10^−3^ M HCl (pH ≈ 2.95), 0.17 mm yr^−1^ in 10^−4^ M HCl (pH ≈ 3.95), and 0.04 mm yr^−1^ in 10^−5^ M HCl (pH ≈ 4.98). The correlation of gravimetrically acquired corrosion data with post-exposure spectrophotometric analysis of fruit juices enabled de-convolution of iron contamination rates from carbon steel corrosion rates in fruit juices. Elemental iron contamination after 50 h of exposure to steel samples was much less than the values predicted from corrosion data (≈40%, 4.02%, 8.37%, and 9.55% for tomato, pineapple, orange, and lemon juices, respectively, relative to expected values from corrosion (weight loss) data). Tomato juice (pH ≈ 4.24) was the least corrosive to carbon steel compared to orange juice (pH ≈ 3.58) and pineapple juice (pH ≈ 3.94). The results confirm that though the fruit juices are acidic, they are generally much less corrosive to carbon steel compared to hydrochloric acid solutions of comparable pH. Differences in the corrosion behaviour of carbon steel in the juices and in the different mineral acid solutions are attributed to differences in the compositions and pH of the test media, the nature of the corrosion products formed, and their dissolution kinetics in the respective media. The observation of corrosion products (iron oxide/hydroxide) in some of the fruit juices (tomato, pineapple, and lemon juices) in the form of apparently hollow microspheres indicates the feasibility of using fruit juices and related wastes as “green solutions” for the room-temperature and hydrothermal synthesis of metal oxide/hydroxide particles.

## 1. Introduction

Most foods and edible materials of natural origin being acidic means that they are prone to be corrosive to metals. Since most foods are acidic [1,2], coupled with their complex and multi-chemical composition, they tend to be corrosive to steels, which are generally active in the acidic pH range. The pH of foods is an important process parameter, as it affects its storage, shelf-life and processing choices [3,4,5]. Cheaper engineering materials such as carbon steels still find use in locally fabricated food processing equipment, mainly in developing countries, and in food canning worldwide. The chemical composition of the metals and more importantly that of the fruit/food processing environment is crucial, as these to a large extent determine the possible corrosion reactions, the corrosion product formed and the possibility of the formation of adherent films or otherwise, thus reducing or enhancing metal loss and entry into the food media [6]. However, the corrosivity of different fruit juices and foods to a metallic material may not be easily predicted based strictly on the pH of the food media. This is due to the possibility that some other chemical components of the food may be capable of influencing either the corrosion process(es) or mechanism(s), and/or the type and surface properties of the final or intermediate corrosion products. Corrosion and corrosion mitigation in fruit and food processing are of vital importance due to the possibility of food contamination and attendant health risks.

The corrosion and degradation of materials result in three major effects—loss of structural integrity or function, reduction in, or loss of, aesthetic quality, and contamination/pollution due to the ingress of materials from the corroding material and/or the escape of contents enclosed by the corroding material into the environment (Figure 1). Though a single corrosion event can manifest all of these effects simultaneously, some effect(s) can be more critical depending on the industry or application. As illustrated in Figure 1, whereas contamination is critical to the food and pharmaceutical industries, structural integrity is the most critical to the construction and aeronautical industries.

Metal corrosion rates in aqueous environments are strongly linked to the stability/solubility of their corrosion products (oxides, hydroxides, oxyhdroxides, etc.), which is in turn dependent on the pH of the medium, as illustrated by the stability diagram for the Fe-H_2_O system [7]. Since fruit juices and foods are acidic and of a complex composition, is it possible to determine metal corrosion and contamination rates on exposure to fruit juices based on corrosion and contamination data from mineral acids of comparable pH? If this is not feasible, can correction factor(s) for these differences be obtained for various foods to aid predictions vital to the design of processing equipment and methods, and packaging materials? How do the corrosion rates of metals in food environments compare with contamination rates? This work is an attempt at finding answers to these questions.

Consequently, in this work, fruit juices are employed as model corrosive media to study the corrosion behaviour of metallic material in food media. This choice is based on their similarities in complex chemical composition and pH with foods, favourable texture, and state (liquid), which makes contiguous contact with metallic test samples easier. This paper, therefore, seeks to study in depth the corrosion of carbon steel in selected fruit juice processing environments (orange juice, pineapple juice, lemon juice, and tomato juice), estimate apparent elemental metal loss into these fruit/food media (contamination) based on corrosion data and the elemental composition of the metal sample, and compare the corrosivity of these juices with that of dilute mineral acid with comparable pH. By adjusting the pH of a mineral acid (HCl) to values consistent with the pH of various fruit juices, the corrosion rates of steel in these test media are compared with the contamination rates (due to the leakage of iron from carbon steel into mineral acid and fruit juices, respectively), and the data compared with respect to pH. The results from this investigation are envisaged to be of importance not only to the food industry but also to various industries/applications (Figure 1) in which the contamination of products due to metallic corrosion is a concern, such as in the chemical processing industry and the health sector (in the corrosion of metallic medical implants).

### 1.1. Complexity of Food (Fruit Juice) Media and Its Implications on Corrosion and Corrosion Inhibition in Food Media

The constituents of fruit and food juices, though multi-component and complex in nature, can, from a corrosion point of view, be broadly classified into three types: (a) those capable of enhancing the corrosion of metals, principal of which are most of the organic acid constituents, (b) those capable of decelerating the corrosion of some metals, such as antioxidants, which are not lacking in most fruits, and (c) those constituents exerting little or no effect on metal corrosion in food fluids. Examples of compounds likely to inhibit corrosion in food media are antioxidants such as ascorbic acid, and even folic acid, which is not an antioxidant [8,9,10], while plausible corrosion enhancers are some of the organic acids present in foods. Some reports [11,12,13] appear to suggest that sulphide compounds in fruits and foods can enhance corrosion. Furthermore, processing methods and the history of fruits can exert influences on the composition of the extracted juice. Vandercook et al. [14] had reported a 64% drop in L-malic acid content and a 34% increase in the total amino acids in extracted lemon juice after the lemon had been stored for 15 weeks. The presence of other food constituents, such as sugars, proteins, pectins, and fatty acids, is reported [15] to inhibit metal (aluminium) corrosion in acidic foods.

It is important to note that some of the organic acids and other chemical compounds present in foods can exert inhibitive effects on the corrosion of certain metals/alloys under acidic conditions consistent with foods such as: citric acid on aluminium in 2 M NaCl solution at pH 2 [16], ascorbic acid on mild steel in 0.01 M H_2_SO_4_ solutions at pH = 2–6 and 0.3% NaCl solution at pH near neutral [9,17], folic acid on mild steel in 0.3% NaCl solution at pH near neutral [9], vitamins on mild steel in acidic media [9,17,18,19,20,21], antioxidant compounds normally present in foods on carbon steel in acidic medium [22,23,24], and amino acids on mild steel in acidic solutions [25,26,27,28,29,30,31,32,33,34,35]. Some peptides (compounds consisting of two or more amino acids) have been reported to bind strongly to aluminium and mild steel [36,37,38,39]. These peptides (such as serine, threonine, and/or histidine [36,37,38,39]) are characteristically rich in hydroxyl-containing amino acids, which are useful in their interactions with metallic surfaces that often result in some corrosion inhibitive effects.

Citric acid, a common component of fruits and fruit juices, has been reported to inhibit the corrosion of aluminium in 2 M NaCl (at pH 2) at concentrations ≤10^−5^ M [16]. Talati and Patel [12] studied the effect of colourants and sweetening agents on copper corrosion in tartaric acid commonly present in various fruits and reported that all the food colourants and sweetening agents studied accelerate the corrosion of copper by tartaric acid. Adewuyi and Oladunjoye [40] studied the effects of sweeteners (glycerol, saccharin, glucose, and sucrose) and colourants (Carmoisine, Sunset Yellow, Poncreau 4R, and Tartazine) on tin-coated steel plates in malic acid and reported that while the sweetening agents inhibited corrosion, the colourants enhanced the corrosion of the tin-plated steels, and attributed the inhibitive effects of the sweeteners to the presence of hydroxyl groups that can form complexes with metal ions, which when insoluble can lead to corrosion inhibition [41,42,43,44,45,46,47,48,49,50].

Furthermore, the action of the varied chemical components in fruit juices and foods in general is bound to be dependent on the metal, and thus more complex with an alloy. For instance, whereas citric acid is reported to inhibit the corrosion of aluminium [16,51,52,53,54], carbon steel [55] and stainless steel [56], it is also reported to enhance the corrosion of tin in similar media [27]. The corrosion scenario in food fluids is made even more complex by the possibility of the synergistic and/or antagonistic activity of the various components and/or between these and dissolved metal ions from alloy corrosion, as in the reported inhibition of carbon steel corrosion by synergy between fructose and the low concentration (50 ppm) of Zn^2+^ in solution [57]. Morad and Hermas [27] studied the effect of some amino acids (25–100 mM glycine, serine, methionine, and cysteine) and vitamin C and some of their binary mixtures on the anodic dissolution of tin in sodium chloride solution and reported a very significant enhancement of the corrosion resistance of tin in the presence of 50–100 mM glycine and methionine, a similar effect in the presence of 100 mM serine, and a reduction in the corrosion resistance of tin in the presence of cysteine and vitamin C. They also reported [27] that whereas tin dissolution in NaCl solution in the absence and presence of glycine, serine, and methionine is a charge transfer controlled process, mixed charge transfer and diffusion control predominates in the presence of cysteine and vitamin C. In the binary mixtures, they reported [27], interestingly, that whereas the corrosion behaviour of tin in the glycine–methionine mixture was similar to effects observed in the presence of the individual components of the mixture, the presence of cysteine in the cysteine–methionine mixture neutralised the inhibitive effects observed in the presence of methionine alone.

It is noteworthy that in spite of its complex chemical composition (being comprised of both corrosion-enhancing and corrosion-inhibiting compounds) and the complicated plausible corrosion scenarios highlighted herein, there have been multiple reports of corrosion inhibition by certain fruit juices, such as: apricot juice on mild steel [58], peach juice on mild steel [59], grapefruit juice (*Citrus paradisi*) on mild steel [60,61], date palm fruit juice on 7075 aluminium alloy [62], aloe vera juice on stainless steel [63], sour cherry juice (*Prunus cerasus*) on St-37 steel in 1 M HCl [64], koehne fruit (*Chaenomeles sinensis* (Thouin)) extract on mild steel in acidic solution [65], watermelon (*Citrullus lanatus* fruit (CLF) extract) on mild steel in 1 M HCl [66], dog-rose (*Rosa canina*) fruit extract on mild steel in 1 M HCl [67], and kiwifruit or Chinese gooseberry (*Actinidia chinensis*) fruit shell extract on mild steel in 1 M HCl [68].

### 1.2. Chemical Composition of Test Media

Since the compositions of foods are very complex, composition information on the fruit juices used in this work is limited to the water and acid contents (including amino acids). For more detailed analysis of the fruit media used, Soucci et al. [69], the Danish Food Database [70], and the USDA Database 2007 [71] are recommended.

#### 1.2.1. Chemical Composition of Lemon Juice

According to the Danish Food Composition Database [70], 100 g of lemon juice (*Citrus limon*) contains 90.3–91.1% moisture, 0.3% ash, and 8.6% total carbohydrates, of which 1.6% are made up of sugars, mainly fructose (0.9%), glucose (0.5%), and saccharose (0.2%). Elemental content includes sodium (6 mg), potassium (160 mg), calcium (110 mg), magnesium (7 mg), phosphorus (21 mg), iron (0.4 mg), copper (0.034–0.069 mg), zinc (0.050–0.30 mg), iodine (0.250–1.50 µg), manganese (0.008 mg), chromium (0.2 µg), selenium (0.120 µg), and nickel (6.66 µg). The acid content of lemon juice is reportedly composed of 46 mg/100 g of L-ascorbic acid (vitamin C).

According to Souci et al., [69] fresh lemon per 100 g is composed of 91 g of water, total nitrogen (0.06 g), protein (0.40 g), fat (0.10 g), available carbohydrate (2.43 g), available organic acids (4.75 g), and minerals (0.34 g). The organic acids present are made up of 4500 mg of citric acid and 250 mg of malic acid. The vitamin/anti-oxidant content is made up of vitamin B1 (40 µg), vitamin B2 (10 µg), nicotinamide (100 µg), panthothenic acid (100 µg), vitamin B6 (52 µg), biotin (300 ng), folic acid (900 ng), and vitamin C (53 mg).

#### 1.2.2. Chemical Composition of Tomato Juice

According to Souci et al., [69] fresh red ripe tomato is composed of 94.20% water. Per 100 g of the dry matter, it is reported to contain 5.66 g citric acid, 0.88 g malic acid, 0.10 g lactic acid, 0.14 g acetic acid, 0.17 g chlorogenic acid, 0.14 g quinic acid, 12.07 mg ferulic acid, 0.03 g fumaric acid, 3.28 mg pyruvic acid, 0.41 g oxaloacetic acid, 2.24 mg salicylic acid, 34.48 mg histamine, 10,206.90 mcg beta-carotene, 3.97 g polyuronic acid, 6.21 g cellulose, and 189.66 mg myoinositol. The amino acid content on the same basis is reported to be 0.10 g tryptophan, 0.40 g threonine, 0.40 g isoleucine, 0.52 g leucine, 0.50 g lysine, 0.12 g methionine, 0.02 g cystine, 0.41 g phenylalanine, 0.21 g tyrosine, 0.40 g valine, 0.31 g arginine, 0.22 g histidine, 0.45 g alanine, 2.09 g aspartic acid, 5.69 g glutamic acid, 0.31 g glycine, 0.28 g proline, and 0.48 g serine. The vitamin and anti-oxidant content per 100 g of dry matter include ascorbic acid (vitamin C) 327.59 mg, thiamin (vitamin B1) 0.98 mg, riboflavin 0.60 mg, pantothenic acid 5.34 mg, vitamin B6 1.72 mg, folic acid 379.31 mcg, beta-carotene 10,206.90 mcg, retinol 1672.41 mcg, alpha-tocopherol (vitamin E) 14.02 mg, tocopherol 13.79 mg, gamma-tocepherol 2.24 mg, phylloquinone (vitamin K) 98.28 mcg, biotin 68.97 mcg, and nicotinamide 9.14 mg [69].

Based on the above information, citric acid is obviously the predominating organic acid in tomato, followed by malic acid. It is reported [72] that processing to juice leads to an increase in the organic acids content in tomato, with acetic acid levels increasing by up to 32%, apparently as a result of the oxidation of aldehydes and alcohols and the deamination of amino acids. Of special interest to this work is the finding of Gould [72] that the relatively high ascorbic acid content in tomatoes favours the presence of iron in its reduced form in this test media, with obvious implications to the corrosion of carbon steel in it. Kader et al. [73] studied the effect of fruit ripeness when harvested on the amino acid composition and flavour of fresh market tomatoes and reported that four amino acids (glutamic acid, γ-aminobutyric acid, glutamine, and aspartic acid) made up ≈80% of the total free amino acids in the tomato fruits.

#### 1.2.3. Chemical Composition of Orange Juice

Orange is made up per 100 g of the edible part of 88.3 g water, 0.11 g total nitrogen, 0.72 g protein, 0.17 g fat, 8.80 g available carbohydrate, 0.45 g dietary fibre, 1.23 g available organic acids, and 0.40 g minerals [69]. The organic acid content is made up of 1112 mg citric acid, 123 mg malic acid and 13 mg of volatile acid. The vitamin/anti-oxidant content is reportedly minute and composed of about 49 mg vitamin C, 41 µg folic acid, 1.4 µg biotin, 50 µg vitamin B6, 230 µg pantothenic acid, 290 µg nicotinamide, 16 µg vitamin B2 and 72 µg vitamin B1 [69]. Karadeniz [74] had reported that the major organic acids of citrus fruits are citric, ascorbic, and malic acids along with much smaller quantities of benzoic, oxalic, and succinic acids. Besides these acids, citrus fruits contain phenolic compounds with reported antioxidant properties [75,76,77], which are likely to exert inhibitive effects on metal corrosion, with the major phenolic compounds in orange juice comprised of phenolic acids and flavanones [78]. Hydroxycinnamic acid and its derivatives, such as caffeic, ferulic, sinapic, p-coumaric, and chlorogenic acids, are reported to be the most important phenolic acids in orange juice [76,79], while the major flavanones in orange juices are hesperidin and narirutin [80], with the flavanones presenting mainly as glycosides [81,82].

#### 1.2.4. Chemical Composition of Pineapple Juice

In total, 100 g of the fresh edible portion of pineapple (*Ananas comosus* (L.) Mer) is reported [83,84] to be generally composed, among other constituents, of 81.2–86.2 g water, 0.005 g oxalic acid, 0.1–0.47 g malic acid, 0.32–1.22 g citric acid, 17–22 μg aminobenzoic acid, 2.5–4.8 μg folic acid, 200–280 μg niacin, 75–163 μg pantothenic acid, 69–125 μg thiamine, 20–88 μg riboflavin, 10–140 μg vitamin B6, 0.002°–0.004 μg vitamin A, and 10–25 μg ascorbic acid. Cárnara et al. [85] reported a ratio close to 2 for citric acid/malic acid content for fresh pineapple juice. Bartolomé et al. [86] investigated the chemical composition of two pineapple cultivars and reported 0.22 to 0.38% malic acid content and 0.8 to 1.27% citric acid content with respect to the fresh weight of the fruits. Tyrosine and tryptophan are reported to be the major amino acids in pineapple [87]; other amino acids in pineapple include asparagine, proline, aspartic acid, serine, glutamic acid, α-alanine, aminobutyric acid, valine, and isoleucine [88].

## 2. Materials and Methods

### 2.1. Test Sample Preparation

Carbon steel rolled sheets (CR1 equivalent to equivalent to EN 1.0338 and AISI 1006) cut into dimensions (1.2 × 20 × 20) were used as gravimetric test samples. The compositions of the carbon steel sheets were determined by optical emission spectroscopy and are presented in Table 1. The carbon sheets used for weight loss were used as received without further polishing. These carbon steel sheets were cut to dimensions of 1.2 × 20 × 20 mm, cleaned in acetone and then in ethanol for 5 min, respectively, in an ultrasonic bath, dried, and weighed in an analytical balance. The initial weights W_1_ of each test coupon were determined to an accuracy of 0.1 mg prior to immersion.

### 2.2. Metallographic Examination

Metallography was employed to obtain microstructural information on the carbon steel sheets used in this study. Metallographic test samples were prepared according to ASTM standard E3-2010 (specimen preparation) [89] and etched in accordance with ASTM standard E407-2007 [90], and the grain sizes were determined according to ASTM standard E112-2010 [91] after etching in 2% Nital. Image acquisition was accomplished using a Nikon Eclipse LV150 optical microscope coupled with a Bresser Mikrocam II 12 MP digital camera.

### 2.3. Test Solution Preparation

The fruit juices were prepared from fresh tomatoes, oranges, and pineapples by mechanical pressing to release the juices. Pulps were removed from the expressed fruit juices by sieving with a fine mesh strainer of mesh size ≈ 250 μm. The respective “filtrates” were used without centrifugation as the respective fruit juice test media. The pH and conductivities of the freshly prepared fruit juices and of the different prepared acid concentrations were determined using a Mettler-Toledo Seven Multi multi-parameter meter.

To study the effect of pH without the influence of the complex composition of the fruit juices, solutions with pH in the range ≈2 to 5 were prepared from hydrochloric acid (37% ACS reagent, Sigma-Aldrich, Saint Louis, MI, USA) and de-ionised water of conductivity ≈18 MΩ (from Barnstead Easypure RF Compact Ultrapure Water System fed with distilled water) to concentrations of 10^−2^, 10^−3^, 10^−4^, and 10^−5^ M HCl to yield approximately 10^−2^, 10^−3^, 10^−4^, and 10^−5^ M H^+^, respectively, without pH adjustment. The calculated and measured pH of these acidic solutions, together with that of freshly prepared fruit juices, are presented in Table 2.

### 2.4. Weight Loss Experiments

Gravimetric tests were done with as-received sheets after preparation as in 2.1 above after total immersion in test solutions at ambient temperature (22 °C) for different time durations (1, 5, 10, 20, 30, 40, and 50 h, respectively). For accuracy, each test was done in triplicate and the average value was used in calculations. The weight loss and the specific weight loss were evaluated using Equations (1) and (2) below, respectively:(1)$$\Delta W\;\;=\;W_{1}-\;W_{2}$$
(2)$$\Delta w=\frac{\Delta W}{A}$$
where Δ*W* is weight loss in mg, *W*_1_ is the initial weight of the sample before immersion, *W*_2_ is the final weight of the sample after immersion, Δ*w* is the specific weight loss of the sample in mg cm^−2^, and *A* is the surface area of the sample in cm^−2^.

Corrosion rate in mm *yr*^−1^ was evaluated from weight loss data using the relation (Equation (3)):(3)$$CR\;\;\left(mm\;yr^{-1}\right)=\;\frac{\Delta W\times \;K}{\rho \;\times A\times \;t}$$
where Δ*W* is weight loss in milligrams, *ρ* is metal density in g cm^−3^, *A* is the area of sample in cm^2^, *t* is exposure time in hours, and *K* is a constant (=87,500).

### 2.5. Scanning Electron Microscopy

Scanning electron microscopy was used to study the surface of the carbon steel sheet before and after 12 h of exposure in the different fruit juice test media. SEM images were acquired using a Hitachi TM4000Plus Tabletop scanning electron microscope with energy dispersive X-ray spectroscopy (EDXS) capability provided by a Bruker Quantax 75 EDS system on CR1 carbon sheet samples as received, and after 12 h of immersion in the different test media.

### 2.6. Atomic Absorption Spectroscopy

A GBC Avanta Atomic Absorption spectrometer was used to determine the Fe contents of the fruit juices and HCl solutions before and after 1 and 50 h of immersion of carbon steel sheets with a surface area of 8.96 cm^2^ in 100 mL of the fruit juices, respectively.

### 2.7. X-ray Diffraction

X-ray diffraction was performed on corrosion products centrifuged from the test media and then dried to a thick paste and on the surface of the steel sheets after immersion using a Rigaku Geigerflex X-ray Powder Diffractometer with Cu-K_α_ radiation at a speed of 3°/min with steps of 0.2° from 10° to 80°.

## 3. Results and Discussion

### 3.1. Compositional and Metallographic Analysis on Metal Samples

The results of the OES compositional analysis (presented in Table 1) show that, besides carbon (0.035%), the other relatively significant alloying elements in the carbon steel that are used are Mn (0.20%), Ni (0.040%), and Cr (0.019%).

The results of the microstructural analysis for the metal sheet CR1 (equivalent to EN 1.0338 and AISI 1006) (Figure 2) reveal a microstructure composed of equiaxed grains of ferrite (light) and pearlite (darker) with ASTM grain sizes evaluated at a magnification of ×200 to be around 9 (ASTM 8.5-9).

### 3.2. Parameters of the Test Solutions

The measured pH and the resistivity of the test solutions (Table 2) show that all the fruit juices had acidic pH, in the range 2.2 to 4.2, and resistivities (2.53 to 3.10 × 10^2^ Ω cm) in the range of that of the 10^−2^ M HCl solution (2.69 × 10^2^ Ω cm) of pH 2.044. Comparison of the resistivities of the test solutions is important, as the resistivity of the electrolyte exerts influences on both the rate [92] and the type of corrosion attack [93]. Confirmation of close similarities in the resistivities of the fruit juices and that of the 10^−2^ M HCl solution indicates insignificant effects due to differences in resistivity and provides a baseline for appreciating the possible effects of resistivity in more dilute HCl solutions.

### 3.3. Results from Weight Loss Measurements

In applications in which metal contamination due to corrosion in food media or leaching into the body due to the corrosion of metallic implants is a concern, the specific weight loss (weight loss per unit of exposed surface) is an important parameter; hence, the weight loss is presented as specific weight loss (Figure 3). In addition, information on the ratio of solution volume to exposed metal surface (specific immersion volume) is important for the application of corrosion data for design purposes. The specific immersion volume used in this work is 11.16 cm^3^ of test media per cm^2^ of metal sample surface. The trend of the results of the gravimetric test (Figure 3a) indicates that tomato juice was least corrosive to carbon steel and that the specific weight losses followed same trend as the pH of the respective fruit juices, increasing with the increasing acidity of the fruit juices. Compared to HCl solutions of comparable pH (Figure 3b), a similar trend was manifested during the first 15 h of immersion.

From the weight-loss data, the corrosion rates were evaluated and presented in Figure 4. From Figure 4, it is observed that the corrosion rates of carbon steel in both the fruit juices followed the same trend of higher corrosion rates in more acidic solutions at the respective sampling times. However, except for lime juice (pH ≈ 2.22) and the 10^−2^ M HCl solution (pH ≈ 2.04) after only 1 h of immersion, at all other sampling times the evaluated corrosion rates of carbon steel in the studied fruit juices were consistently higher than those of HCl solutions of comparable pH.

### 3.4. Results from Scanning Electron Microscopy

Figure 5 presents the scanning electron images of carbon steel (CR1 ≈ EN 1.0338) surfaces after 12 h of immersion in the studied fruit juices and in the respective HCl solutions for comparison. Significant corrosion products were observed on carbon steel surfaces exposed to tomato juice (pH ≈ 4.24) (Figure 5a), pineapple juice (pH ≈ 3.94) (Figure 5c), orange juice (pH ≈ 3.54) (Figure 5e), and even on the surface of carbon steel exposed to lemon juice (at pH ≈ 2.22) (Figure 5g). The SEM images of carbon steel immersed in the respective HCl solutions; 10^−5^ M HCl (pH = 4.98) (Figure 5b), 10^−4^ M HCl (pH = 3.95) (Figure 5d), and 10^−3^ M HCl (pH = 2.95) (Figure 5f) reveal the presence of pits and corrosion products. However, in spite of the presence of pits, the quantities of corrosion products observed in these solutions were significantly less than those observed in fruit juices with a similar pH. In 10^−2^ M HCl (pH = 2.04) (Figure 5h), large pits were observed but very little corrosion products were detected in and around the pits. This is attributed to the fast dissolution of corrosion products formed on carbon steel in this solution. Considering that the pH of lemon juice (pH ≈ 2.22) and 10^−2^ M HCl (pH ≈ 2.04) are quite similar and the most acidic in their respective categories, the presence of pronounced pits in both is attributed to similarities in the corrosion mechanism in both solutions, which suggests that pH is a significant factor in carbon steel corrosion in fruit juices and HCl solutions and increases in prominence with the lowering of solution pH. In contrast, the dearth of corrosion products on the surfaces of carbon steel samples exposed to lemon juice (pH ≈ 2.22) and 10^−2^ M HCl (pH ≈ 2.04), which is much more severe in 10^−2^ M HCl, is attributed to a pH-enhanced increase in the dissolution kinetics of corrosion products (iron oxides/hydroxides) in these two low-pH test media. The presence of comparatively more corrosion products on carbon steel immersed in lemon juice (pH ≈ 2.22), and generally in the other three fruit juices compared to HCl solutions of comparable pH, is indicative of some inhibitive effects to iron oxide dissolution in the studied fruit juices attributable to their complex chemical constitution. Interestingly, the corrosion products formed in the fruit juices (particularly pineapple, tomato, and orange juices) present as (apparently hollow) microspheres. This indicates that the complex chemical composition of the fruit juices exerts influences on both the kinetics of oxide dissolution and the morphology of the oxides formed in them. It is worth noting that the resolution of the microspheres on the carbon steel surfaces was achieved by using the surface sensitive observation mode using a 5 kV acceleration voltage, which by significantly reducing the interaction volume of the incident electrons gave prominence to surface features. Earlier SEM observations using a 15 kV accelerating voltage (see Appendix A) permitted an easy observation of pits but failed to detect the presence of the iron oxide microspheres.

Energy dispersive X-ray spectroscopy (EDXS) was carried out on selected areas and points on the surfaces of carbon steel samples immersed in the different test media and some of the data presented as elemental maps (Figure 6) for carbon steel in tomato juice, pineapple juice (Figure 7), orange juice (Figure 8), and lemon juice (Figure 9), respectively. The corresponding respective spectra are presented in the accompanying Appendix A. The results from the point analysis (not shown) and elemental mapping (Figure 6, Figure 7, Figure 8 and Figure 9) indicate that in all the media tested, post-exposure carbon steel surfaces were covered with a thin oxide layer even when “an aggregation” of corrosion products is not observed. Hence, it is obvious that the corrosion of carbon in the fruit juices and HCl solutions manifest as uniform and pitting corrosion. This similarity in the presentation of carbon steel surfaces after exposure to fruit juices and HCl solutions is deemed to be indicative of similarities in the corrosion mechanism in both types of test media.

In order to obtain further insight into the nature of pits observed in SEM examinations of sample surfaces (Figure 5), cross-sectional SEM images of carbon steel samples after 12 h’ exposure to the respective test media were acquired, and the results for samples immersed in 10^−3^ M HCl and in tomato juice are presented in Figure 10 and Figure 11, respectively.

The cross-sectional SEM image with elemental maps for carbon steel sheet surfaces after 12 h of immersion in 10^−3^ M HCl solution (pH = 2.95) presented in Figure 10 shows higher counts for oxygen and lower counts for iron compared to intact carbon steel surface and interior. This observation confirms the presence of oxide-filled pits.

The cross-sectional SEM image with elemental maps for carbon steel sheet surfaces after 12 h of immersion in tomato juice presented in Figure 11 shows a pit with some iron oxide(s) near the bottom of the pit and close to the mouth of the pit. Based on observations from scanning electron microscopy and EDS mapping of both the carbon steel surfaces (Figure 5, Figure 6, Figure 7, Figure 8 and Figure 9) and cross-sections (Figure 10 and Figure 11) after exposure to the respective test media employed in this work, it is concluded that the corrosion of carbon steel in fruit juices and acidic chloride solution manifest as a combination of uniform and pitting corrosion. Although carbon steel is unable to form the type of tenacious passive films observed in stainless steels, due to its nil or insignificant content of elements such as Cr, Mo, and Ni, the formation of passive film(s) on carbon steel [94,95,96] in a variety of alkaline solutions has been reported. The observed incidence of pitting corrosion in carbon steel samples exposed to fruit juices and HCl solutions in this work is consistent with the earlier reports [97,98,99,100,101] of pitting corrosion in carbon steel exposed to acidic solutions.

### 3.5. Results from X-ray Diffraction

The results from the X-ray diffraction of corrosion products formed on the carbon steel sheets (Figure 12) show that in orange, lemon, and pineapple juices, maghemite (γ-Fe_2_O_3_) and akaganéite (Fe^8+3^(O,OH)_16_Cl_1.3_) were the dual corrosion products, while maghemite (γ-Fe_2_O_3_), akaganéite (Fe^8+3^(O,OH)_16_Cl_1.3_), and magnetite (Fe_3_O_4_) were the corrosion products detected on carbon steel exposed to tomato juice. Since akaganéite is linked to steel corrosion in the presence of a plentiful supply of both Cl^−^ and Fe^2+^ [102], it is inferred that in all the fruit juices, sufficiently high Cl^−^ and Fe^2+^ concentrations were most likely generated and sustained, thus favouring the formation of akaganéite.

The superior corrosion resistance of carbon steel in tomato juice might be attributed to the unique presence of magnetite (Fe_3_O_4_) as a corrosion product on the carbon steel samples exposed to tomato juice, in addition to akaganéite and maghemite also observed on samples exposed to other fruit juices. When present as a corrosion product, magnetite is known to form a magnetite layer on the inner surface of carbon steels [103,104,105,106,107,108]. The semiconducting properties of iron oxide electrodes have been reported [109]. Magnetite can be semi-conductive in certain conditions with a small bandgap of the order of 0.1 eV [110,111,112]. In addition, the semiconducting property of maghemite detected in carbon steel samples exposed to each of the fruit juices is well known [113,114]. Surface oxides present on corroding metal surfaces are known to influence corrosion processes particularly when they are semiconducting [115]. Sato [115] had postulated that p-type surface oxides accelerate metal corrosion by raising the electrode potential (shift in the anodic direction), while n-type surface oxides decrease metallic corrosion by lowering the electrode potential (shifts of the corrosion potential towards the cathodic direction) based on the reports by earlier workers [116,117,118,119,120] of similar potential shifts and reduced corrosion rates in metallic copper and stainless steels samples in contact with η-type titanium oxides.

### 3.6. Results from Atomic Absorption Spectrometry

The results from the atomic absorption spectrometric analysis of fruit juices exposed to the same surface area of carbon steel for the same duration (Table 3) do not follow the trend established by the gravimetric test results.

This discrepancy might be due to the fact that, whereas gravimetric tests give information on metal loss (oxidized metal) from an exposed sample, atomic absorption spectrometry (AAS) gives information on the quantity of metal that actually passed into the food media and thus some idea as to the nature (e.g., solubility) of the corrosion products formed in each media. This position is corroborated by the SEM images of test samples after immersion in the test media (Figure 5), in which the presence and abundance of corrosion products on the metal surface followed this trend: pineapple juice > tomato juice > orange juice > lemon juice > 10^−2^ M HCl (in which visible corrosion products were scarcely observed on the metal surface). The trend of metal loss into the test media as seen from AAS results (Table 3) was tomato juice (2765.4 ppm) > lemon Juice (2095.3 ppm) > orange juice (1503.7) > pineapple juice (452.7), after subtracting background iron content. It was noted that tomato juice formed a greater variety of corrosion products on the sample (Figure 10) and the lowest corrosion rates (Figure 3 and Figure 4), but it had the most iron content in the test media (Table 3). This observation can be partly attributed to the formation of a more soluble phase in the corrosion products formed in tomato juice (most probably magnetite from the XRD results of Figure 12). The fast saturation of iron in the test volume could hinder the dissolution of more corrosion products and/or cause a subsequent change in dissolution kinetics that favour the formation of insoluble products upon saturation. Dissolution rates of various iron oxides and hydroxides in acidic media and iron-leaching rates from these oxides and hydroxides have been reported to vary with the type of iron oxide or oxy-hydroxide [121,122,123,124,125]. Sidhu et al. [121] studied the dissolution rates per unit of surface area of various iron oxides in acidic media (0.5 M HCl) at 25 °C and, by analysis of the linear regions of their respective dissolution curves, ranked their dissolution rates in the following order: lepidocrocite > magnetite > akaganéite > maghemite > hematite > goethite. Such a sequence in the dissolution rates of iron oxides/hydroxides was attributed to differences in the chemical composition and crystal structure. From this ranking, faster dissolution kinetics for magnetite detected only on carbon steel sample immersed in tomato juice compared to both akaganéite and maghemite is confirmed. Though this enhanced dissolution kinetics of magnetite detected only on carbon steel samples exposed to tomato juice can account for the higher concentrations of iron in tomato juice, it does not account for the observation of more corrosion products in carbon steel exposed to tomato juice. A plausible explanation can be inferred from reports that indicate that in addition to manifest differences in iron oxide/oxy-hydroxide dissolution rates, iron leaching rates from these iron oxides and hydroxides also vary with the type of oxide or oxy-hydroxide, with magnetite presenting high iron leaching rates [121,122,123,124,125].

### 3.7. Estimation of Apparent Elemental Metal Loss and Correlation of Weight Loss to Elemental Contamination of Fruit Juices

In food environments, contamination issues can be of greater importance than equipment failure(s), which can be handled by over-design. The possibility of estimating contamination levels into the food being processed from corrosion data can be desirable. With information on the chemical composition of a metal sample and either corrosion rate, weight loss, or specific weight loss data, the quantity of each element that might possibly contaminate the test media can in principle be calculated. Using the weight loss data obtained after 50 h of immersion and the spectrometrically determined iron concentrations in the respective fruit juices (Table 3), and the volume of test solution (100 mL in both weight loss and AAS tests), the weight of iron (contamination) in the respective fruit juices was determined. By assuming that the constituent elements of the carbon steel sample (in Table 1) enter into solution in proportion to their concentrations in the carbon steel sample, a correction was made to these data, and an estimate of the proportion of the measured weight loss attributable to iron dissolution was obtained. The results obtained by this treatment and correlated to the spectrometrically determined concentration/mass of iron in the respective fruit juices are presented in Table 4.

From Table 4, it can be observed that significantly less iron entered the test solution than was expected from the weight loss data, even after correcting for its concentration (99.674 wt. %) in the carbon steel. The highest elemental iron contamination relative to weight loss data was ≈ 40% in tomato juice and less than 10% in the other fruit juices studied.

Such de-convolution of corrosion data and elemental contamination data in fluids is vital in applications where contamination is a primary concern, such as food processing, studies on metallic implants in the body, and related applications (Figure 1). The results from this work (Table 4) demonstrate that corrosion rate(s) does not implicitly translate to contamination rates and is not equivalent to contamination rates.

### 3.8. Correlation of Corrosion Rate of Carbon Steel in Fruit Juice of Varying pH with Acidic Chloride Media of Varying pH

Since most fruit juices and many food media are acidic, and corrosion data were acquired for carbon steel in fruit juices and in HCl solutions of comparable pH, an attempt was made to correlate the corrosion rates of carbon steels in fruit juices and in HCl solutions of comparable pH with respect to pH, and the results are presented in Figure 13 as a plot of corrosion rate versus test media pH after 5 h’ immersion. From Figure 13, it is observed that irrespective of the test method at pH values between 4 and 2.5, the fruit juices manifest corrosion rates that are higher than those of HCl solutions of comparable pH.

The correlation of the corrosion rates of carbon steel in the fruit juices with respect to their pH and in acidic chloride media of varying pH (different concentrations of HCl) presented in Figure 13 indicates that, with respect to pH, the fruit juices were generally less corrosive to carbon steel than acidic chloride media of comparable pH. Linear regression indicates that, whereas the corrosion rate of carbon steel in the fruit juices varied linearly with pH (Equation (4)), that of acidic chloride solution exhibited a power relationship with pH (Equation (5)).
(4)$$CR=-0.1308pH+0.5859{\;}^{\;\;\;\;\;\;}\left(R^{2\;\;\;\;\;\;}=0.95\right)$$
(5)$$CR=46.835pH^{-5.176\;\;\;\;\;\;}\;\;\;\;\;\;\left(R^{2\;\;\;\;\;\;}=0.98\right)$$

Laque and Copson [126] had stated that fruit acids present in foods are apparently more corrosive than would be expected from their hydrogen ion concentration, but the result from this work, while not strictly disproving their statement, demonstrates that whole fruit juices are generally less corrosive than would be expected from their hydrogen ion concentration (pH values). Hence, it is posited herein that corrosion data from metal exposure to inorganic acids of comparable pH generally underestimate the corrosion rates of carbon steel in fruit juices; thus, they might not be quite as useful in the design of equipment for handling fruit juices of similar pH.

## 4. Conclusions

The use of fruit juices as model corrosive media to study the corrosion of metallic material in foods has been demonstrated. This treatment can be applied in scenarios in which contamination due to material degradation is a concern. The corrosion of carbon steel in selected fruit juices and in mineral acids of comparable pH as the fruit juices was studied based on data from gravimetric measurements. The corrosion of carbon steel in fruit juices and HCl solutions present as a combination of uniform and pitting corrosion. Tomato juice (pH ≈ 4.24) was the least corrosive to carbon steel compared to orange juice (pH ≈ 3.58), pineapple juice (pH ≈ 3.94), and lemon Juice (pH ≈ 2.22). The results confirm that though the fruit juices are acidic, they are generally much less corrosive to carbon steel compared to hydrochloric acid solutions of comparable pH. Consequently, it is difficult to make accurate predictions of the corrosion rate of carbon steel in acidic fruit juices on the basis of their pH. In spite of this, for design purposes, consideration of the corrosion rates of carbon steels obtained in mineral acids of comparable pH can be helpful, as this has been observed to generally over-estimate the corrosion rate of carbon steel in the fruit juices in the pH range ≥2.6 to 4.24. Differences in the corrosion behaviour of carbon steel in the juices, and in the different mineral acid solutions, are attributed to differences in the compositions and pH of the test media, the nature of the corrosion products formed and their dissolution kinetics in the respective media. The dominant effect is attributed to slower iron oxide dissolution kinetics plausibly linked to the robust chemical compositions of fruit juices. The correlation of gravimetrically determined corrosion data with data from the analysis of test media enabled the de-convolution of elemental iron contamination rates from carbon steel corrosion rates in the fruit juices; thus, it was possible to demonstrate that corrosion rate(s) does not implicitly translate to contamination rates and is not equivalent to contamination rates. Elemental iron contamination of the fruit juices (vol. = 100 mL) after 50 h’ exposure to carbon steel samples with a surface area of 8.96 cm^2^ showed that elemental iron contamination was much less than what might have been assumed from corrosion data (≈ 40% for tomato juice, 4.02% for pineapple juice, 8.37% for orange juice, and 9.55% for lemon juice, relative to expected values from corrosion (weight loss) data). X-ray diffraction showed that maghemite (γ-Fe_2_O_3_) and akaganéite (Fe^8+3^(O,OH)_16_Cl_1.3_) were the dual corrosion products in lemon, pineapple, and orange juices, while magnetite (Fe_3_O_4_) in addition to these was detected in tomato juice. The detection of corrosion products (iron oxide/hydroxide) on carbon steel samples immersed in some of the fruit juices (tomato, pineapple, and lemon juices) in the form of apparently hollow microspheres may have potential technological importance, as it indicates the feasibility of using fruit juices and related wastes as “green solutions” for the room-temperature and hydrothermal synthesis of metal oxide/hydroxide particles.

## Figures and Tables

**Figure 1 materials-14-04755-f001:**
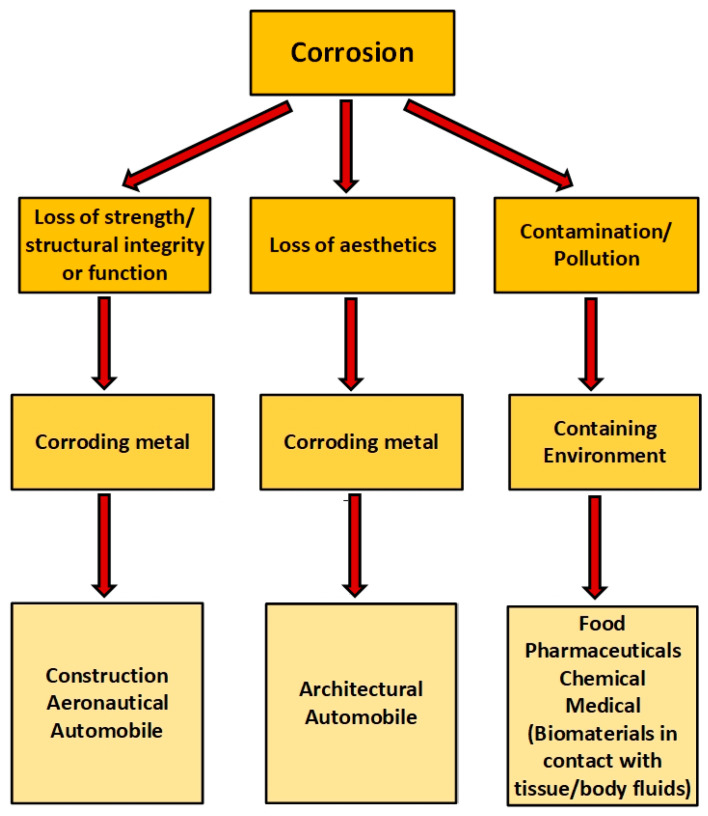
Illustration of the three major effects of corrosion and relevance to different industries.

**Figure 2 materials-14-04755-f002:**
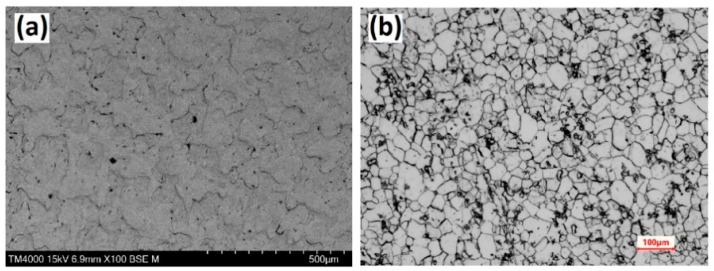
(**a**) SEM image of as-received carbon steel sheets; (**b**) microstructural (optical) image of CR1 Carbon Steel Sheet after etching in 2% Nital.

**Figure 3 materials-14-04755-f003:**
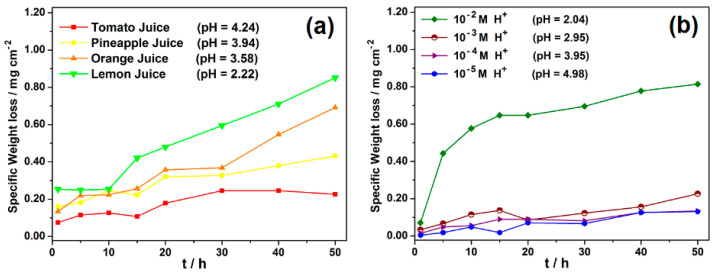
Plot of specific weight loss as a function of immersion time, respectively, for carbon steel in (**a**) different fruit juices and (**b**) HCl solutions of comparable pH.

**Figure 4 materials-14-04755-f004:**
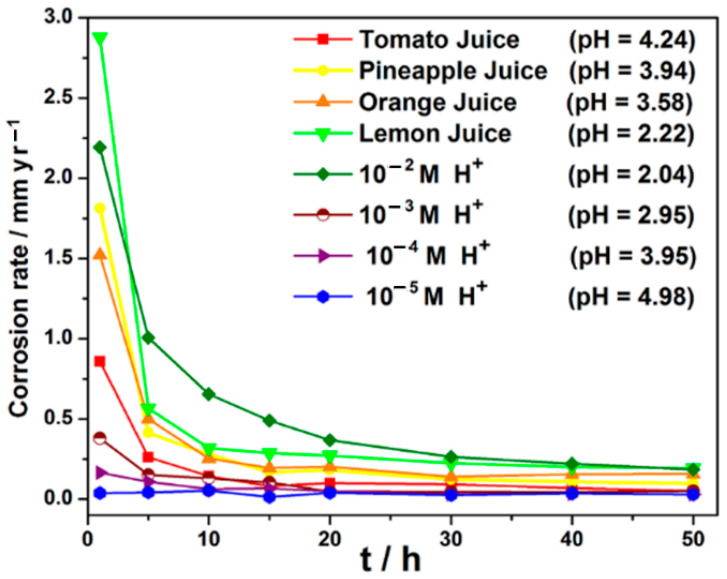
Plot of corrosion rate as a function of immersion time, respectively, for carbon steel in different fruit juices and HCl solutions of comparable pH from gravimetric data.

**Figure 5 materials-14-04755-f005:**
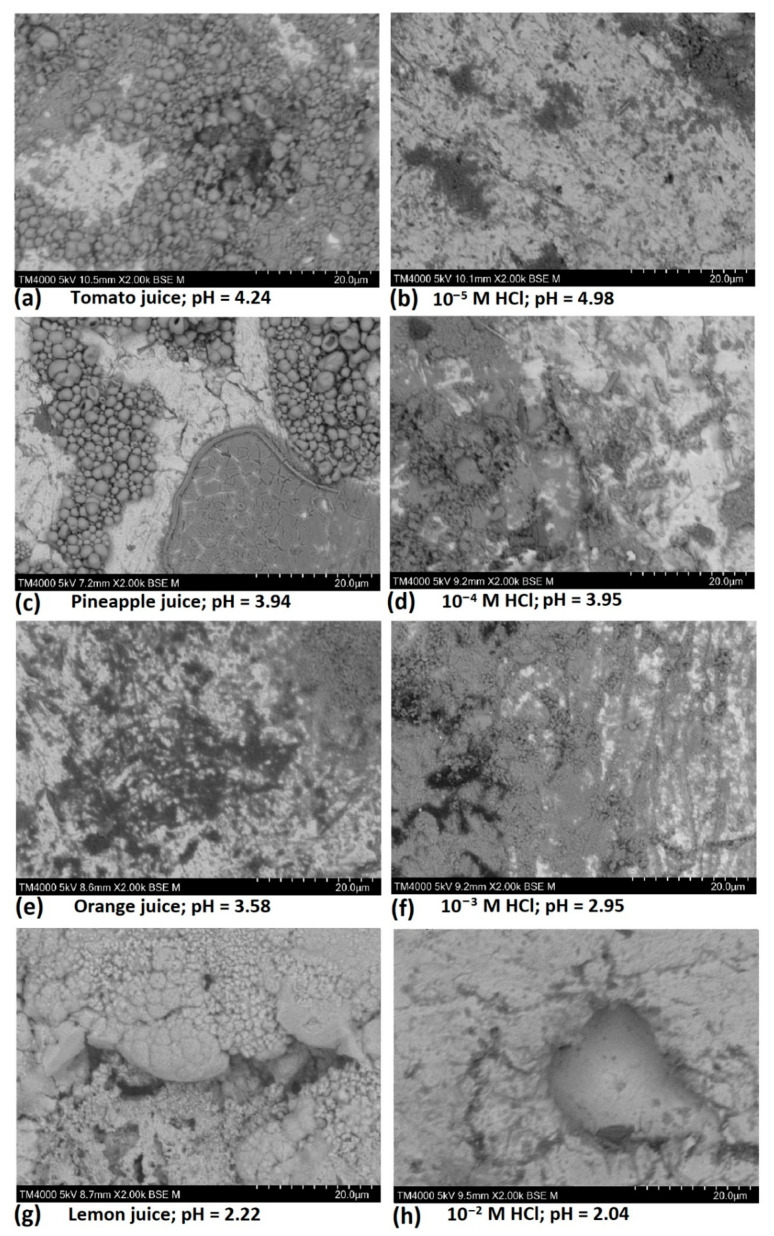
SEM images of carbon steel sheet (CR1 ≈ EN 1.0338) surfaces after immersion in (**a**) tomato juice, (**b**) 10^−5^ M HCl, (**c**) pineapple juice, (**d**) 10^−4^ M HCl, (**e**) orange juice, (**f**) 10^−3^ M HCl, (**g**) lemon juice, and (**h**) 10^−2^ M HCl, respectively.

**Figure 6 materials-14-04755-f006:**
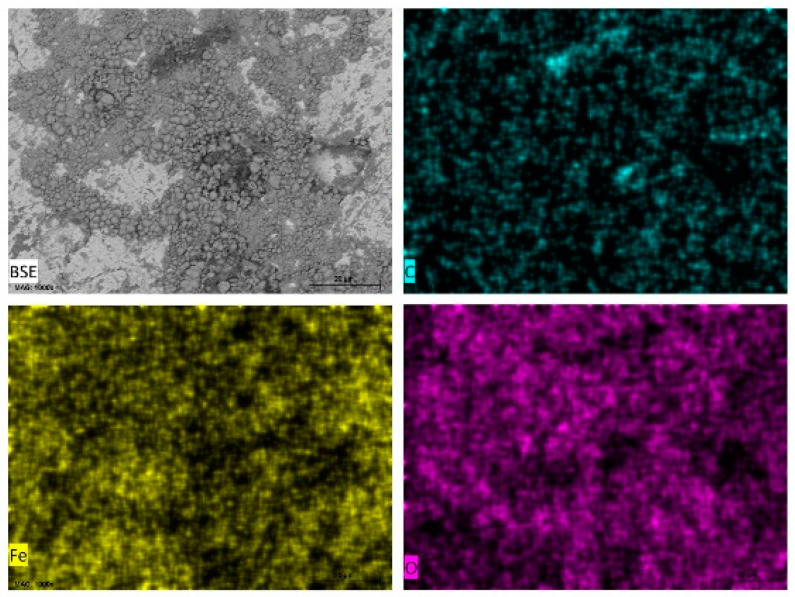
SEM/EDS mapping images of carbon steel sheet (CR1 ≈ EN 1.0338) surfaces after immersion in tomato juice (pH = 4.24).

**Figure 7 materials-14-04755-f007:**
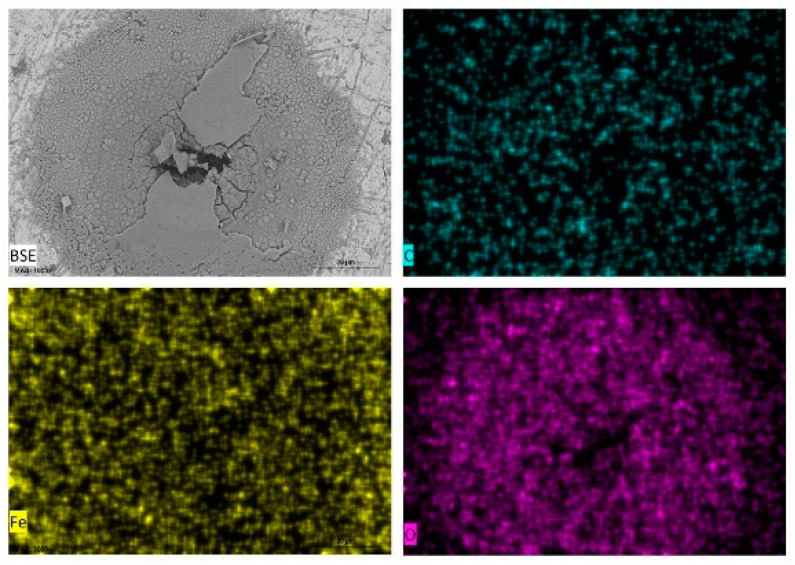
SEM/EDS mapping images of carbon steel sheet (CR1 ≈ EN 1.0338) surfaces after immersion in pineapple juice (pH = 3.94).

**Figure 8 materials-14-04755-f008:**
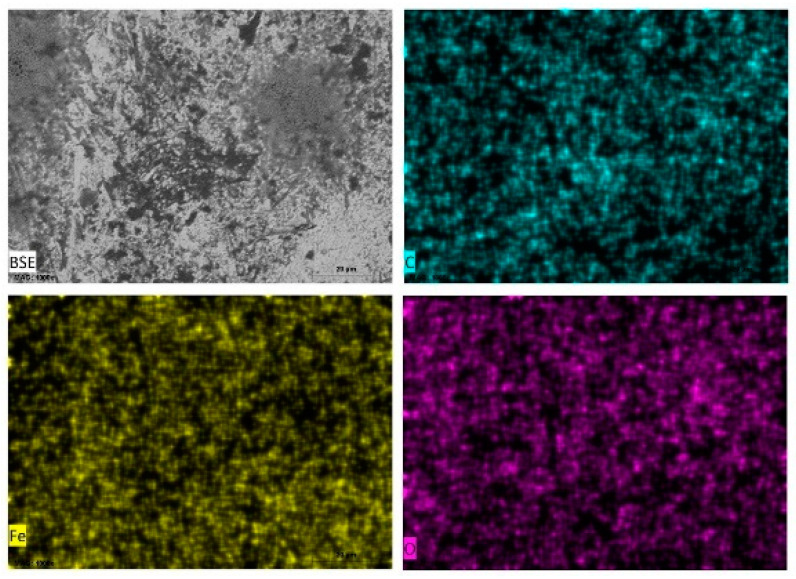
SEM/EDS mapping images of carbon steel sheet (CR1 ≈ EN 1.0338) surfaces after immersion in orange juice (pH = 3.58).

**Figure 9 materials-14-04755-f009:**
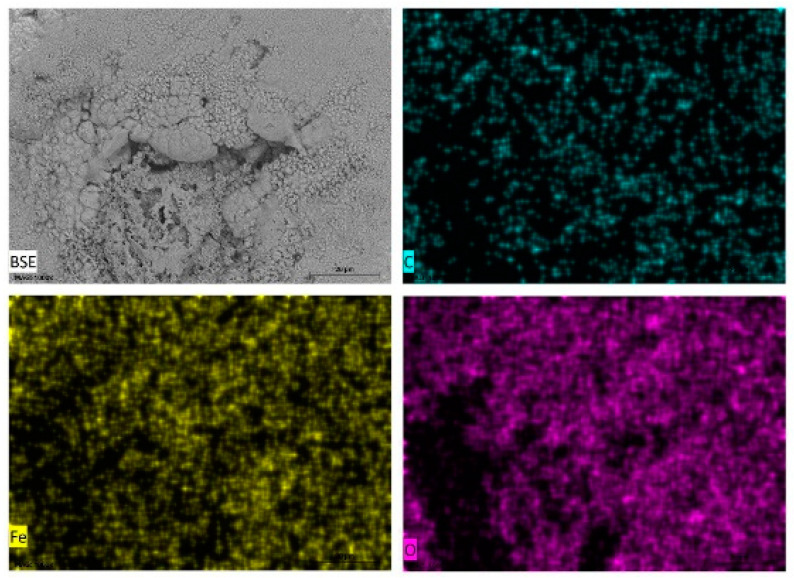
SEM/EDS mapping images of carbon steel sheet (CR1 ≈ EN 1.0338) surfaces after immersion in lemon juice (pH = 2.22).

**Figure 10 materials-14-04755-f010:**
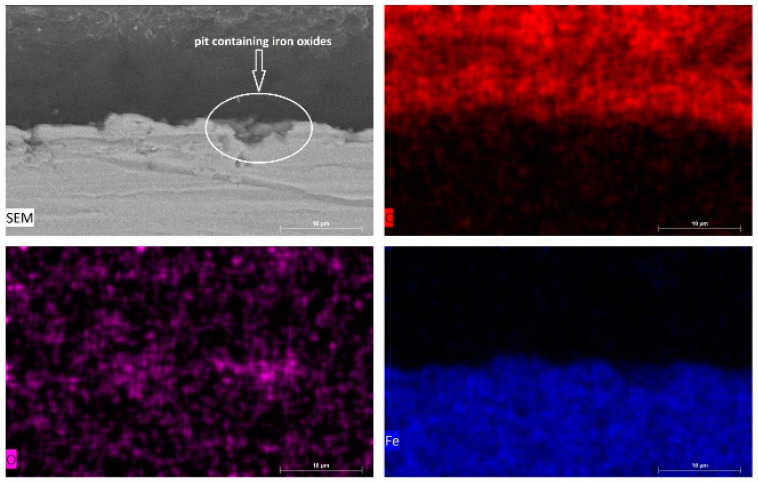
Cross-sectional SEM image with elemental maps for carbon steel sheet surfaces after 12 h of immersion in 10^−3^ M HCl solution (pH = 2.95).

**Figure 11 materials-14-04755-f011:**
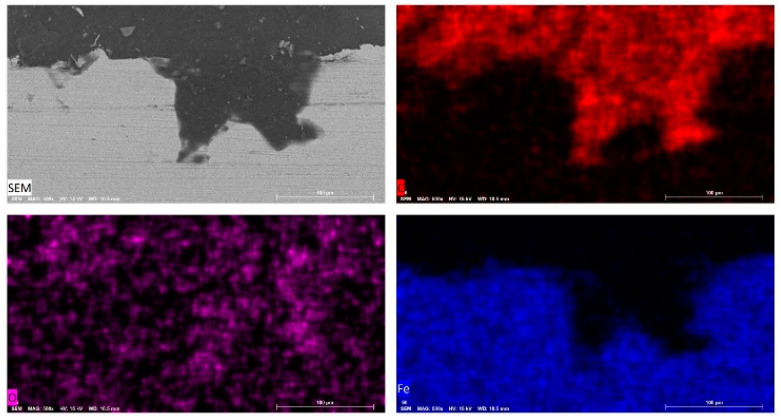
Cross-sectional SEM image with elemental maps for carbon steel sheet surfaces after 12 h of immersion in tomato juice (pH = 4.24).

**Figure 12 materials-14-04755-f012:**
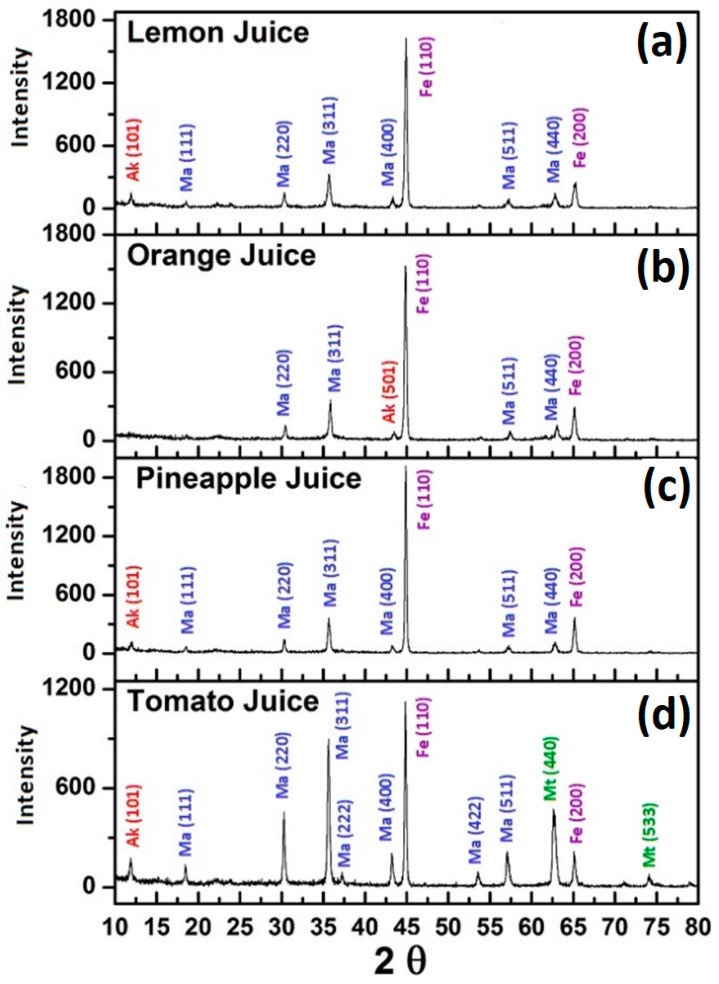
X-ray diffraction patterns acquired from carbon steel (CR1 ≈ EN 1.0338) surfaces after 12 h of immersion in (**a**) lemon juice, (**b**) orange juice, (**c**) pineapple juice, and (**d**) tomato juice, (Ak is Akaganéite, Ma is Maghemite, and Mt is Magnetite).

**Figure 13 materials-14-04755-f013:**
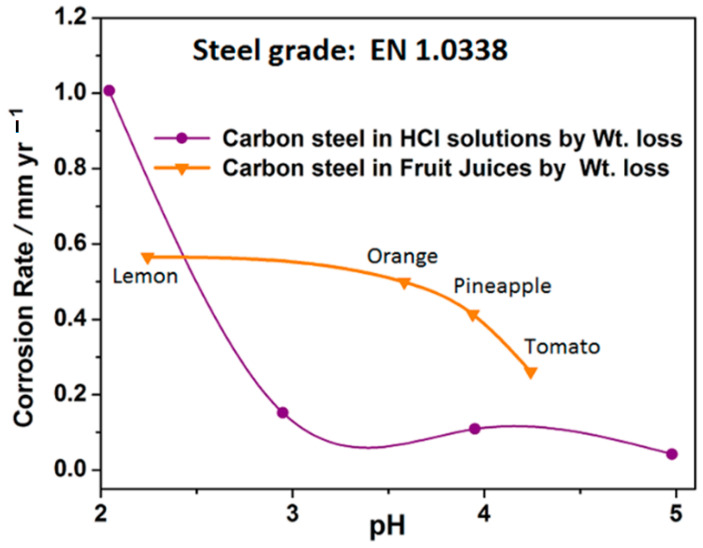
Plot of corrosion rate vs. test media pH for carbon steels after 5 h immersion in fruit juices, and acidic chloride solutions of varying pH evaluated from gravimetric data.

**Table 1 materials-14-04755-t001:** Composition of Carbon Steel Sheet.

Carbon Steel Description	% C	% Si	% Mn	% P	% Ni	% Cr	% S	% Pb	% Fe
Sheet CR1	0.035	0.010	0.20	0.014	0.040	0.019	0.006	0.002	99.674

**Table 2 materials-14-04755-t002:** Measured and calculated parameters (pH and resistivity) of test solutions.

Test Media	pH (Calc.)	pH (Measured)	Resistivity (Ω-cm)
Tomato	-	4.242	3.04 × 10^2^
Orange	-	3.582	3.10 × 10^2^
Lemon	-	2.224	2.53 × 10^2^
Pineapple	-	3.940	4.57 × 10^2^
10^−2^ M HCl	2	2.044	2.69 × 10^2^
10^−3^ M HCl	3	2.950	1.96 × 10^3^
10^−4^ M HCl	4	3.952	2.18 × 10^4^
10^−5^ M HCl	5	4.979	2.46 × 10^5^

**Table 3 materials-14-04755-t003:** Atomic absorption spectrometry results of Fe content in fruit juices before and after 50 h of immersion of carbon steel samples with a surface area of 8.96 cm^2^.

Sample	Fe Content in Fresh Juice (ppm)	Fe Content after 50 h of Carbon Steel (8.96 cm^2^) Immersion (ppm)
Tomato Juice	1.066	2766.5
Pineapple Juice	0.745	453.4
Orange Juice	0.881	1504.6
Lemon Juice	0.936	2096.2

**Table 4 materials-14-04755-t004:** Correlation of weight loss and atomic absorption spectrometry results of Fe content in fruit juices after 50 hours’ immersion of carbon steel samples with a surface area of 8.96 cm^2^ in 100 mL of test media.

Test Media	Wt. Loss of Carbon Steel after 50 h Immersion in (mg)	Wt. Loss Attributable to Fe (mg) *	Fe Content from AAS after 50 h of Carbon Steel Immersion (ppm)	Molarity of Fe in Test Media from AAS (mol L^−1^) **	Fe Content per Litre after 50 h of Carbon Steel Immersion (mg)	Fe Content in 100 mL of Solution (mg)	% of Wt. Loss Attributable to Fe Manifesting in Solution (%)
Tomato Juice	0.7	0.697718	2766.5	0.0495	2.7643	0.276433	39.62
Pineapple Juice	1.13	1.1263162	453.4	0.0081	0.4523445	0.04523445	4.02
Orange Juice	1.8	1.794132	1504.6	0.0269	1.5022	0.150223	8.37
Lemon Juice	2.2	2.192828	2096.2	0.0375	2.0942	0.2094188	9.55

***** Based on Fe content in carbon steel (99.674 wt. %). ** Based on atomic weight of Fe (55.845 g mol^−1^).

## Data Availability

The raw/processed data required to reproduce these findings cannot be shared at this time, as the data also form part of an ongoing study.

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
