# Peer review of "Comparative Gravimetric Studies on Carbon Steel Corrosion in Selected Fruit Juices and Acidic Chloride Media (HCl) at Different pH"

_materials, 2021, doi:10.3390/ma14164755_

Round 1

Reviewer 1 Report

This work was focus on the comparative gravimetric studies on carbon steel corrosion in selected fruit juices and acidic chloride media (HCl) at different pH. The results of this manuscript are reasonably interesting. In my viewpoint, I recommend this manuscript to be accepted for publication after a minor revision. Some suggestions for the authors to improve the manuscript:

  1. At present, the melt of ready-to-eat food is generally used in aluminum cans. In this article, what are the main starting points for selecting a carbon steel used for engineering materials as the food preparation material, and what are the existing advantages?
  2. Please unify the formula format and picture ruler style.
  3. Please check out the English grammar carefully.
  4. Please check out the references carefully.
  5. Some references need be refreshed. And some recent references need be cited.

Author Response

Reviewer 1 This work was focus on the comparative gravimetric studies on carbon steel corrosion in selected fruit juices and acidic chloride media (HCl) at different pH. The results of this manuscript are reasonably interesting. In my viewpoint, I recommend this manuscript to be accepted for publication after a minor revision. Some suggestions for the authors to improve the manuscript: Reviewer 1, Comment 1: At present, the melt of ready-to-eat food is generally used in aluminum cans. In this article, what are the main starting points for selecting a carbon steel used for engineering materials as the food preparation material, and what are the existing advantages? Response to Reviewer 1, Comment 1: This is true. However, the author has observed the used of low-grade steels in fabrication of food processing equipment in developing countries due to the lower cost. However, similar studies with aluminium and tin are on-going and will be reported subsequently as these 2 metals are more often in contact with packaged foods; Al in aluminium foils or cans, and Sn in tin-coated steel cans. Insights from the current study would be valuable to similar investigations involving other metals. Moreover, in tin-coated steel cans if the tin-coating is breached, foods can come into contact with the underlying steel. Reviewer 1, Comment 2: Please unify the formula format and picture ruler style. Response to Reviewer 1, Comment 2: Differences in the picture ruler style are due to acquisition from different equipment (optical microscope and scanning electron microscope) that offer different style options. Reviewer 1, Comment 3: Please check out the English grammar carefully. Response to Reviewer 1, Comment 3: English grammar checks have been made. Reviewer 1, Comment 4: Please check out the references carefully. Some references need be refreshed. And some recent references need be cited. Response to Reviewer 1, Comment 4: Some of the dated references are due to citation of literature that first expressed the ideas. Some recent references have now been added.

Reviewer 2 Report

The submitted manuscript provides good coverage of the topic of corrosion in the food industry. The presented results demonstrate well the corrosion study, including microscopy, chemical surface, and corrosion rate analysis. 
However, the manuscript has several points for improvement/clarification:
1. The authors stated that carbon steel (CS) exposed to acidic pH juice and HCl had pit formation which is related to localized damage. However, it is known that CS does not contain elements such as Cr, Ni, Mo in sufficient amounts to be exposed to localized corrosion in such acidic environments stated in the manuscript. This is since, at low pH, the domination of H+ takes place where H+ will be reduced to H2 and CS will be dissolved producing Fe2+ into the solution. And as juice contains water, the water will reduce OH- ions resulting in Fe-based corrosion product formation which is rust. Therefore, rust can not be in such condition considered as a passive film that after its breakdown can lead to localized corrosion, i.e. pittings. Therefore, the question arose why the authors lead the discussion about localized corrosion but not uniform/general corrosion for CS in the stated environment?
2. The authors provided the readers with the SEM (and EDS) top surface morphology of the exposed samples, but not crossed sectioned SEM/EDS results. Based on the first point above, the cross-section would show if the corrosion layer formed on CS led to pit formation or not.
3. The comparative analysis of corrosion for CS in juices and HCl requires additional explanation since in HCl solution the content of Cl- ions will lead to much aggressive solution formation where the corrsoion rate of CS may be higher than in juice. Please, clarify this point.

The submitted manuscript is to be revised and submitted for the second round of peer review.

Author Response

Responses to Reviewer 2

Reviewer 2

The submitted manuscript provides good coverage of the topic of corrosion in the food industry. The presented results demonstrate well the corrosion study, including microscopy, chemical surface, and corrosion rate analysis.

However, the manuscript has several points for improvement/clarification:

Reviewer 2, Comment 1:

The authors stated that carbon steel (CS) exposed to acidic pH juice and HCl had pit formation which is related to localized damage. However, it is known that CS does not contain elements such as Cr, Ni, Mo in sufficient amounts to be exposed to localized corrosion in such acidic environments stated in the manuscript. This is since, at low pH, the domination of H+ takes place where H+ will be reduced to H2 and CS will be dissolved producing Fe2+ into the solution. And as juice contains water, the water will reduce OH- ions resulting in Fe-based corrosion product formation which is rust. Therefore, rust can not be in such condition considered as a passive film that after its breakdown can lead to localized corrosion, i.e. pittings. Therefore, the question arose why the authors lead the discussion about localized corrosion but not uniform/general corrosion for CS in the stated environment?

Response to Reviewer 2, Comment 1:

 I appreciate and understand the reviewer’s position on the report of incidences of pitting in the acidic pH ranges used in this work considering that the carbon steel used in this study contains no or trace quantities of passive film “promoters” in steel like Cr, Ni, and Mo. However, careful and repeated observations of samples used in this study indicates that the observed pits on the carbon steel surface are not artefacts. In this updated version of the manuscript elemental maps of carbon steel cross -sections post-exposure has now been included. These cross-sectional maps confirm the presence of empty pits (in regions low on oxygen and iron) and oxide filled pits in (regions rich in oxygen). It appears that the formation of a tenacious passive layer as found in stainless steels is not a criterion for the formation of pits. The presence of corrosion products/film(s) that affect mass transport of species in a corrosion cell appears to be sufficient to support pitting in the systems studied.

In addition, literature indicates that this is not an isolated observations as earlier workers have reported this phenomenon in carbon steel in acidic pH range [Silva, V. M., & Williams, L. F. (1977). Pitting of plain carbon steels in acidic solution. Surface Technology, 6(2), pp. 131-137. https://doi.org/10.1016/0376-4583(77)90003-6  ; Noor, E. A., & Al-Moubaraki, A. H. (2008). Corrosion behavior of mild steel in hydrochloric acid solutions. Int. J. Electrochem. Sci, 3(1), 806-818. ; Guo, P., La Plante, E. C., Wang, B., Chen, X., Balonis, M., Bauchy, M., & Sant, G. (2018). Direct observation of pitting corrosion evolutions on carbon steel surfaces at the nano-to-micro-scales. Scientific reports, 8(1), 1-12. https://doi.org/10.1038/s41598-018-26340-5  ; Alves, V. A., & Brett, C. M. (2002). Influence of alloying on the passive behaviour of steels in bicarbonate medium. Corrosion science, 44(9), 1949-1965. https://doi.org/10.1016/S0010-938X(02)00019-7  ; Kolawole, S. K., Kolawole, F. O., Enegela, O. P., Adewoye, O. O., Soboyejo, A. B. O., & Soboyejo, W. O. (2016). Pitting corrosion of a low carbon steel in corrosive environments: experiments and models. In Advanced Materials Research (Vol. 1132, pp. 349-365). Trans Tech Publications Ltd.  https://doi.org/10.4028/www.scientific.net/AMR.1132.349 ] .

However, I fully agree with the esteemed reviewer on the electrochemical mechanism of the corrosion process. This aspect  is not reported herein and will be the subject of an up-coming communication.

Reviewer 2, Comment 2:

The authors provided the readers with the SEM (and EDS) top surface morphology of the exposed samples, but not crossed sectioned SEM/EDS results. Based on the first point above, the cross-section would show if the corrosion layer formed on CS led to pit formation or not.

Response to Reviewer 2, Comment 2:

Cross-sectional SEM images with elemental maps have now been added. These confirm the presence of both empty pits and pits containing iron oxides.

Reviewer 2, Comment 3:

The comparative analysis of corrosion for CS in juices and HCl requires additional explanation since in HCl solution the content of Cl- ions will lead to much aggressive solution formation where the corrosion rate of CS may be higher than in juice. Please, clarify this point.

Response to Reviewer 2, Comment 3:

An idea is to use this aggressive media to get insight into the worst-case scenarios for corrosion of metallic materials in these fruit juices and other food media.

Reviewer 2, Comment 4:

The submitted manuscript is to be revised and submitted for the second round of peer review.

Response to Reviewer 2, Comment 4:

The manuscript has now been revised in line with earlier reviewers’ comments and additional data included.

Round 2

Reviewer 2 Report

The author has improved the manuscript. It is a great revision. I agree with the corrections and additions. Accepted for publication.